



# A Highly Sensitive and Selective Laser-Based BTEX Sensor for Occupational and Environmental Monitoring

Mhanna Mhanna[1], Mohamed Sy[1], Ayman Arfaj[2], Jose Llamas[2], Aamir Farooq[1]

[1]Mechanical Engineering Program, Physical Science and Engineering Division, Clean Combustion Research Center, King
Abdullah University of Science and Technology (KAUST), Thuwal 23955-6900, Saudi Arabia
[2]Saudi Aramco, Environmental Protection, 9f Al-Midra Tower, Dhahran 31311, Saudi Arabia

*Correspondence to*: Aamir Farooq (aamir.farooq@kaust.edu.sa)

**Abstract.** A mid-infrared laser-based sensor is designed and demonstrated for trace detection of benzene, toluene, ethylbenzene, and xylene isomers at ambient conditions. The sensor is based on a distributed feedback inter-band cascade laser emitting near 3.29 μm and an off-axis cavity-enhanced absorption spectroscopy configuration with an optical gain of ∼2800. Wavelength tuning and a deep neural networks (DNN) model were employed to enable simultaneous and selective BTEX measurements. The sensor performance was demonstrated by measuring BTEX mole fractions in various mixtures. At an integration time of 10 seconds, minimum detection limits of 11.4, 9.7, 9.1, 10, 15.6, and 12.9 ppb were achieved for benzene, toluene, ethylbenzene, m-xylene, o-xylene, and p-xylene, respectively. The sensor can be used to detect tiny BTEX leaks in petrochemical facilities and to monitor air quality in residential and industrial areas for workplace pollution.

## 1 Introduction

Human activity, especially transportation and industrial emissions, have deleterious impacts on air quality. Petrochemical industries are the major emitters of volatile organic compounds (VOCs) (Yassaa et al.; Zhang et al.). In particular, benzene, toluene, ethylbenzene and xylene isomers (BTEX) are emitted from engine exhausts, gasoline service stations, petrochemical refineries, and the paint and rubber industries (Foo). Indoor sources of BTEX include gas boilers (Helmer), oil stoves (Greenburg et al.), candles and cleaning products (Sekar et al.).

The ubiquity and adverse effects of these species, even at the parts per billion (ppb) level, on environmental and human health have attracted increasing attention. Studies have strongly linked global warming, stratospheric ozone depletion, and petrochemical smog to BTEX emissions (Sinha et al.). The carcinogenic and mutagenic effects of BTEX species on human health are recognized by the World Health Organization (WHO) (Organization) and US Environmental Protection Agency (EPA) (Williams et al.), and have raised tremendous concerns in recent years.

Exposure to high concentration of volatile organic compounds (VOCs) may cause acute symptoms such as irritations of the nose, throat, and eyes, in addition to causing headache, nausea, dizziness, and allergic skin reactions. Moreover, some components of VOCs such as BTEX can lead to chronic health risks. Benzene is a carcinogen, and toluene exposure can lead to devastating neurological disorders (Dorsey). Long-term exposure to xylene may cause headache, extreme tiredness, tremors,





impaired concentration, and short-term memory loss. If high concentrations of hydrocarbons replace oxygen in breath, it can sensitize the heart to stress hormones causing abnormal rhythms and ventricular fibrillation that can cause sudden death (Adgey et al.). A brief exposure (< 30 seconds) to high concentrations of BTEX (> 2%) and low-oxygen atmosphere can result in rapid onset of respiratory depression, hypoxia, and fatal cardiac arrhythmias (Miller and Mazur). Long-term BTEX exposure can

have narcotic effects, causing dizziness, rapid disorientation, and confusion that can lead to loss of judgment, narcosis, and incapacitation (Sugie et al.).

A multitude of studies have proved a strong association between occupational exposure to benzene by inhalation and an increased incidence of certain types of leukemia. Adverse health effects of occupational benzene exposure include acute myelogenous leukemia, blood diseases, plastic anemia, injury of the immune system, and menstruation disorders (Esteve-

Turrillas et al.). Occupational toluene exposure is associated with central nervous system depression, tiredness, dizziness, faintness, memory loss, nausea, appetence decrease, hearing loss, and unconsciousness. Acute occupational exposure to elevated levels of ethylbenzene may result in neurological effects such as light-headiness, dizziness and eye irritation. Chronic occupational exposure to xylene may affect the central nervous system, cause skin irritation, skin stimulation, dryness, skin rapture, blister and skin dermatitis (Bina et al.).

Reliable sensors are extremely important for these harmful pollutants (Mhanna et al., 2023). Most existing sensors are based on sample extraction and offline analysis in the laboratory. For instance, gas chromatography, mass spectrometry and Fourier transform infrared spectroscopy (FTIR) (Roubaud et al.; Gregory et al.) involve highly expensive and complex instrumentation, require trained analysts for operation and are not suitable for field application due to their size and weight. Flame- and photo- ionization detection (FID and PID) are commonly used for benzene measurement, but are affected by high

humidity. In addition, these sensors are time consuming due to the intensive sample collection, transportation, and analysis steps, which leads to the loss of spatial and temporal resolution of the measurements.

In recent years, advanced laser sources have been developed, which possess narrow emission linewidths and broad frequency tuning that provide access to the absorption bands of various molecules. In addition, significant improvements have been achieved in the robustness, stability, power consumption and heat management of these lasers. These characteristics make

spectroscopic absorption sensors simple and accurate, and thus open a new strategy in remote sensing of gases. The ultraviolet (UV) wavelength region is an attractive option for BTEX measurements. Tunnicliff et al. demonstrated the measurement of BTEX components based on absorption photometry in the UV range circa 1950 (Bui and Hauser). However, the broad absorption features of many other hydrocarbons in that range do not permit interference-free measurements. Alternatively, infrared (IR) absorption spectra of hydrocarbons provide the possibility of more selective sensing of gaseous species

(Elkhazraji et al., 2023; Elkhazraji et al., 2022). Numerous sensors for hydrocarbons using the mid-IR range have appeared in literature, and some devices have been commercialized (Farooq et al.). Previously, Sydoryk et al. (Sydoryk et al.) developed a mid-IR laser-based sensor for BTX detection near 1038 cm$^{-1}$. However, this region contains significant interference from ozone and ethylene, and even trace quantities of these interfering gases can result in large errors in the measured concentrations of BTEX.



The similar absorbance spectra of BTEX species in the IR are the main cause of the poor selectivity of existing sensors. In a previous work, we demonstrated selective BTEX sensing using deep neural networks (DNNs) to differentiate the similar absorbance spectra of BTEX molecules (Mhanna et al.). In this work, cavity-enhanced absorption spectroscopy (CEAS) coupled with an updated DNN model are used to achieve high sensitivity and selectivity for a miniaturized BTEX sensor for occupational/industrial hygiene and atmospheric monitoring.

## 2 Methodology

### 2.1 Beer-Lambert law

The principle of laser absorption spectroscopy relates the spectral absorbance to the gas properties via Beer-Lambert Law (Swinehart):

$$\alpha_{SP} = \sigma(\nu, T, P) \cdot n(T, P) \cdot L \cdot \chi \tag{1}$$

where $\alpha_{SP}$ is the measured absorbance for a single pass of the laser, $\sigma(\nu, T, P)$ is the temperature- and pressure-dependent absorption cross-section that describes species propensity to absorb light at a frequency $\nu$, $n$ is the total number density of the gas mixture, $L$ is the physical absorption path-length, and $\chi$ is the mole fraction of the absorbing species. At frequencies where multiple species absorb, a composite absorbance results from the summation of individual absorbances. For a given absorption cross-section and experimental conditions $(T, P)$, smaller values of mole fraction $(\chi)$ can only be measured by increasing the

path-length, $(L)$. One effective methodology for increasing $L$ utilizes two highly reflective parallel concave mirrors to create an optical cavity through multiple reflections of the laser light.

     Cavity ringdown spectroscopy (CRDS) and cavity-enhanced absorption spectroscopy (CEAS) have been used for trace detection of gaseous species (Kosterev and Tittel). In CRDS, laser intensity accumulates inside the cavity and then its characteristic exponential decay time is measured, while in CEAS, the laser wavelength is typically tuned over a specific

spectral range to record spectrally-resolved absorption features. The laser intensity is attenuated by absorbance of the target species and the characteristics of the cavity mirrors. The cavity absorbance, $\alpha_{CEAS}$, is related to an equivalent single-pass absorbance, $\alpha_{SP}$, via Eq. (2) (Nasir and Farooq):

$$\alpha_{SP} = -ln\left(1 - \frac{e^{\alpha_{CEAS}-1}}{G}\right) \tag{2}$$

where $\alpha_{CEAS}$ is measured from the transmitted $(I_t)$ and incident $(I_0)$ laser intensities using:

$$\alpha_{CEAS} = -ln\left(\frac{I_t}{I_0}\right) \tag{3}$$

and $G$ is the cavity gain, which is related to the mirrors' reflectivity (R) by:

$$G = \frac{R}{1-R} \tag{4}$$



Here, $\alpha_{SP}$ is calculated from Eq. (2) and then used in Beer-Lambert relation (1) to calculate the target species mole fraction. By combining Eqs. (1) to (4) for the case of a weak absorber and high reflectivity ($\alpha_{SP} \to 0$ and $R \to 1$), and letting $U = 1 - e^{-\alpha_{SP}}$, we get:

$$\alpha_{CEAS} = G \cdot U \tag{5}$$

## 2.2 Wavelength selection

BTEX and many other hydrocarbons have broad absorbance spectra in the ultraviolet (UV) range, which obfuscates interference-free and selective sensing. Alternatively, the infrared (IR) region provides opportunities for highly selective detection of hydrocarbons and air pollutants. Based on the IR absorbance spectra of BTEX shown in Fig. 1, given in the Pacific Northwest National Laboratory (PNNL) database (Sharpe et al.), the best frequency to measure BTEX is near 700 cm⁻¹. However, this spectral region is currently not widely accessible by commercially available semiconductor lasers. Benzene measurements near 3090 cm⁻¹ (Sur et al.) and 1040 cm⁻¹ (Lewicki et al.) suffer significant interference from isoprene and ozone, respectively. Therefore, we selected the ro-vibrational band of BTEX near 3040 cm⁻¹, shown in the inset of Fig. 1, which has negligible isoprene and ozone interference (Mhanna et al., 2022a). This region contains broad and overlapping absorbance spectra of BTEX which make it challenging to separate the individual absorbance contributions. We note that the selected region is insensitive to the presence of carbon dioxide, water vapor, acetone, ammonia, nitrous oxide, carbonyl sulfide, formic acid, nitrogen dioxide and nitric acid (Sharpe et al.; Gordon et al.).

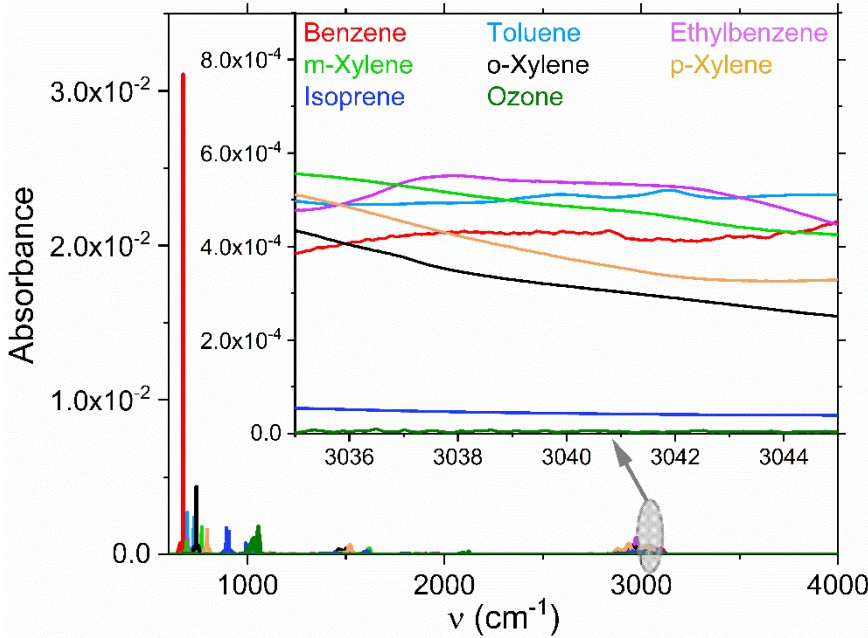

**Figure 1: BTEX spectra in the C-H stretching band. Spectral data taken from the PNNL database (Sharpe et al.) over 600 – 4000 cm⁻¹ and simulated at T = 25 °C, P = 1 atm, L = 30 cm, χ = 1000 ppm. The inset shows a close-up view over 3035 – 3045 cm⁻¹.**



## 2.3 Deep neural networks

Being inspired by the biological systems of neurons and synapses , artificial neural networks (ANNs) are supervised learning
algorithms that mimic human brain (Ma et al.). ANN architecture is based on an input layer which handles the input tensor,
hidden layers which contain neurons, followed by an output layer which handles the target tensor (Agatonovic-Kustrin and
Beresford). Deeper architectures which comprise more than a single hidden layer are referred to as deep neural networks
(DNNs). DNN models are typically developed using a fully connected multilayer perceptron (MLP) associated with a
backpropagation algorithm (Hornik et al.; Nielsen). The back propagation of neural networks (BPNN) involves a feed forward
stage to train the input tensor, followed by a calculation and backpropagation of the resulting error, and finally updating the
weights in order to minimize the error (Werbos). A fixed learning rate and partial derivatives are used to prevent
under/overfitting in back propagation (Guo et al.).

## 2.4 Sensor setup

A distributed feedback interband cascade laser (DFB-ICL, Nanoplus) was utilized to emit near 3.29 µm with an output power
~ 2 mW. Higher power commercial lasers are not available in this region. To facilitate the optical alignment, the ICL was co-
aligned with a 670 nm red laser (Thorlabs). An optical cavity was formed utilizing two ZnSe mirrors of 99.97% nominal
reflectivity (LohnStar Optics) in a 5 cm gas sampling cell with an off-axis alignment. This suppressed the spurious coupling
noise significantly in comparison to an on-axis cavity. The laser wavelength was tuned over 3039.25 – 3040.5 cm$^{-1}$ by a linear
ramp of the laser injection current at 1 kHz scan rate. A 7.62-cm Germanium Fabry-Pérot etalon was used to convert the scan
time to frequency (wavenumbers). The transmitted laser intensity was collected via a focusing lens on to an AC-coupled, TE-
cooled photodetector (Bandwidth of 1.5 MHz, Vigo Systems). Figure 2 shows the schematic of the sensor setup. All
measurements were carried out at 298 K and 1 atm.



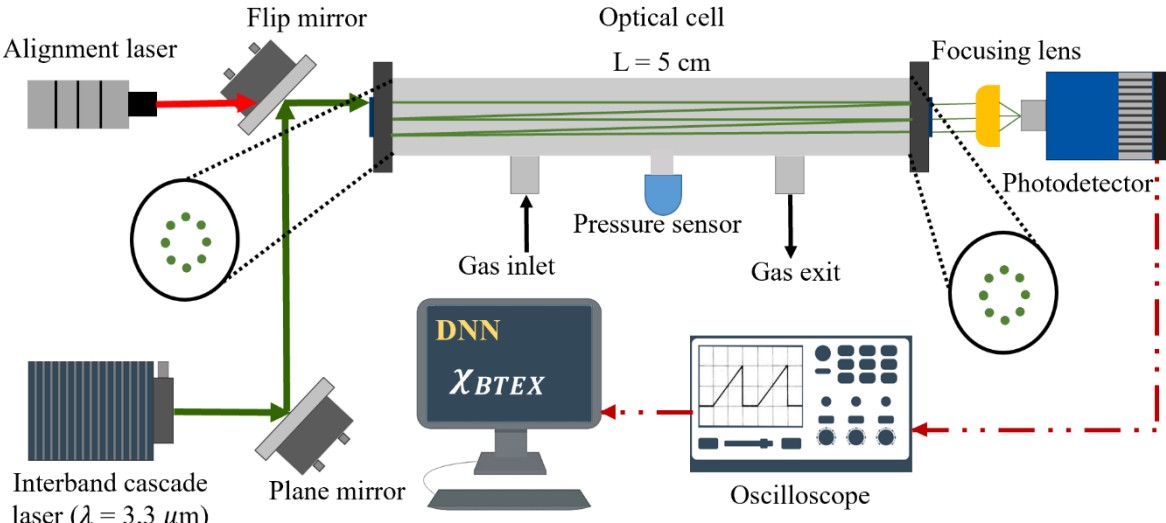

**Figure 2: Optical schematic of the sensor.**

## 3 Sensor calibration

### 3.1 Allan variance

To characterize the long-term stability and obtain the optimum integration time of the sensor, Allan variance was calculated as follows (Giglio et al.):

$$s_x^2(t) = \frac{1}{K} \sum_{k=1}^{K} \frac{1}{2} (x_{k+1} - x_k)^2 \qquad (6)$$

where $K$ is the total number of data-points, $x_k$ is the $k$th data-point averaged over an integration time $t$, and $(x_{k+1} - x_k)$ is the difference between adjacent values of $x_k$. The Allan variance plot of the obtained absorbance signal of the sensor is shown in Fig. 3. The BTEX mixture contains 231, 379, 168, 182, 99, and 113 ppb of high-purity benzene (99.8%, Sigma-Aldrich), toluene (98%, VWR Chemicals), ethylbenzene (99.8%, Acros Organics), m-xylene, o-xylene, and p-xylene (99.8%, Acros Organics), respectively, diluted in nitrogen (99.999 %). Allan variance decreases at early times of the measurement due to the domination of thermal noise until reaching its minimum value at around 10 seconds (Allan), and later increases due to laser intensity fluctuations (Giglio et al.).




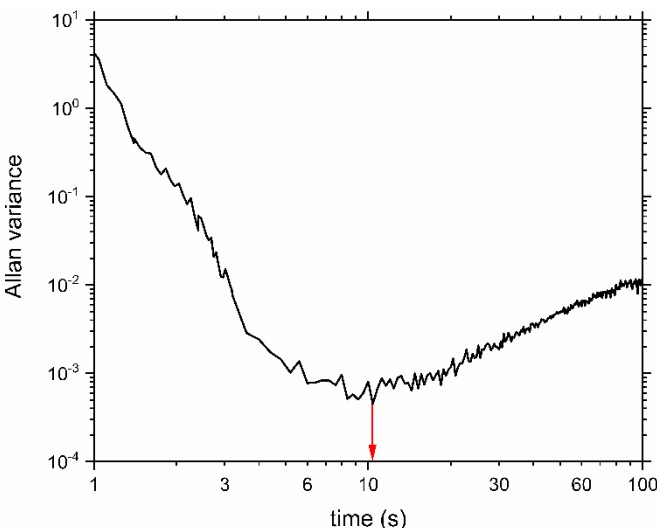

**Figure 3: Allan variance plot of the BTEX sensor. Optimal integration time is 10 seconds.**

**3.2 Cavity gain evaluation**

Cavity gain is determined by the mirrors' reflectivity, as shown in Eq. (4). Typically, significant deviation is observed between the actual and the manufacturer's nominal reflectivity specification, resulting in large uncertainty in the cavity gain. Thus, a more precise value of the cavity gain ($G$) needs to be measured. We performed experiments for the evaluation of $G$ using various mixtures of each BTEX species in nitrogen with known compositions (30 – 2000 ppb). Based on Eq. (5), $G$ was determined by plotting $\alpha_{CEAS}$ against $U$, as shown in Fig. 4. Here, $G$ comes out to be $2789 \pm 80$, which translates into mirrors'

reflectivity $R = 99.964 \pm 0.001\%$. The gain uncertainty ($\pm 80$) was determined based on a 95% confidence interval of the measured values. This value of cavity gain increases the effective laser path-length from 5 cm (physical length of the gas sampling cell) to 139 m. All measurements reported here were carried out at ambient conditions with temperature and pressure uncertainties of 0.03% and 0.12% of the readings, respectively. Absorption cross-sections are reported to have 1.66% uncertainty (Sharpe et al.) and the physical path-length is known with an uncertainty of < 1%. Uncertainties of $\alpha_{CEAS}$ and $U$

are calculated using the following two equations and come out to be 2.1% and 2.9%, respectively.

$$\frac{\partial \alpha_{CEAS}}{\alpha_{CEAS}} = \left[ \left( \frac{\partial I_0}{I_0} \right)^2 + \left( \frac{\partial I_t}{I_t} \right)^2 \right]^{\frac{1}{2}}$$

(7)

$$\frac{\partial U}{U} = \left[ \left( \frac{\partial \sigma}{\sigma} \right)^2 + \left( \frac{\partial P}{P} \right)^2 + \left( \frac{\partial \chi}{\chi} \right)^2 + \left( \frac{\partial T}{T} \right)^2 + \left( \frac{\partial L}{L} \right)^2 \right]^{\frac{1}{2}}$$

(8)





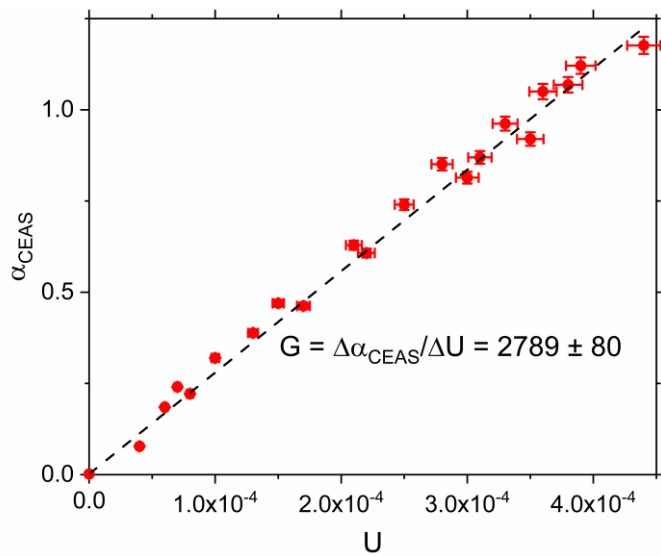

**Figure 4: Cavity absorbance as a function of U (see Eq. 5). The slope represents the cavity gain. Measurements were carried out over T = 298 K and P = 1 atm. BTEX mole fractions were varied over 30 – 2000 ppb.**

### 3.3 Free spectral range

The laser spot size ($s$) in the cavity for laser wavelength $\lambda$ of 3.29 $\mu m$, mirrors' radius of curvature $R_m$ of 1 m, and physical pathlength $L$ of 5 cm was calculated to be 1.89 mm following the procedure described in our previous work (Mhanna et al., 2020). The rotation of the spots on the mirror surface after one round trip is an angle $\theta$ given by:

$$\theta = cos^{-1}\left(1 - \frac{L}{R_m}\right) = 31° \qquad (9)$$

Thus, at least 11 non-overlapping spots would be formed on the mirrors. Although the cavity mirrors had a diameter of 25.4 mm, the exposed diameter of the mirrors was limited to about 8 mm due to the geometry of the mirror mounts. Therefore, the Lissajous circular spot pattern was aligned within a diameter of about 7 mm to avoid the light clipping near the edges of the mounts. This resulted in a circumference of about 22 mm which fits about 11 non-overlapping laser spots, and thus agrees with the spot rotation calculations.

The effective free spectral range ($FSR_{eff}$) is calculated via Eq. (10) (Sayres et al.):

$$FSR_{eff} = \frac{c}{2nL} \qquad (10)$$

where $c$ is the speed of light in air, $n$ is the number of non-overlapping spots, and $L$ is the physical length of the optical cell. Due to the 11 non-overlapping spots, $FSR_{eff}$ was reduced from 3000 MHz in an on-axis alignment to 273 MHz in the off-axis alignment, which significantly suppressed the cavity coupling noise by increasing the density of the cavity mode (Engel et al.).



# 4 Sensor performance

## 4.1 DNN prediction model

Python 3.8 software was utilized to build the deep neural network (DNN) model, which was shown in a previous work (Mhanna et al.) to outperform other prediction models in such classification applications due to its flexibility of tuning hyper-parameters.

The model input is the composite BTEX absorbance spectra and the output is the individual BTEX contributions to the composite absorbance. A total of 1000 simulated and 100 measured BTEX absorbance spectra were utilized over the frequency range of 3039.25-3040.5 cm$^{-1}$. Simulated spectra were generated using Eq. (11):

$$A = \sum_1^4 a_k * A_k \tag{11}$$

where $A$ is the composite absorption spectrum, $a_k$ is the individual contribution of species $k$, and $A_k$ is the reference absorption

spectrum of species $k$ (Sharpe et al.). To highlight the weak absorbance features of BTEX species, a min-max normalization is applied to scale the data:

$$A_{normalized} = \frac{A - A_{min}}{A_{max} - A_{min}} \tag{12}$$

where $A_{normalized}$ is the normalized absorbance, and $A_{max}$ and $A_{min}$ are the maximum and minimum absorbance values in $A$, respectively. Normalized absorbance is fed to the DNN model, where the input tensor consists of the 1100 scaled spectra, and

the target tensor is the contribution of each BTEX species to the composite absorbance. The data are randomly split into 80/20 train/test datasets. A random search algorithm (RandomSearchCV) was applied to tune the hyper-parameters, including the number of nodes and hidden layers, activation functions, dropout layer, weight regularization, learning rate, momentum, number of epochs and batch size. The DNN model was optimized based on lowest root-mean-squared-error (RMSE), resulting in four hidden layers with 132, 64, 32, and 16 nodes, respectively. ReLU and Adamax (learning rate = 0.001 and momentum

= 0.9) were opted as the activation function and optimizer, respectively. The model was run on 2000 epochs with a batch size of 64, and the validation loss was monitored with a patience of 20 epochs to avoid overfitting. The algorithm flowchart is shown in Fig. 5.

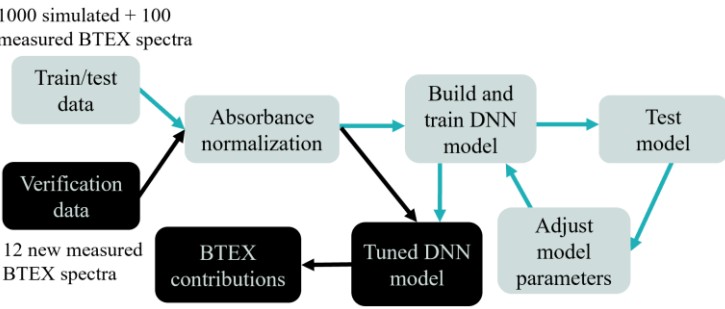

**Figure 5: Schematic for the overall process to build the DNN model and to extract BTEX contributions. Blue arrows correspond to**
**model tuning using the 1000 simulated and 100 measured spectra, while black arrows correspond to extracting BTEX coefficients**
**from the 12 separately measured spectra.**



## 4.2 Sensor validation

Gas samples containing BTEX species were manometrically prepared and DNN was applied to obtain the mole fractions of target species from the composite measured absorbance spectra. In these mixtures, the mole fraction of each BTEX species

was varied over 0 – 2000 ppb, and measurements were carried out in two optical cells of 50 cm and 5 cm optical lengths. As a representative case, Figure 6 shows the measured absorbance spectrum of a mixture containing 121, 278, 422, 0, 151, and 775 ppm of benzene, toluene, ethylbenzene, m-xylene, o-xylene, and p-xylene, respectively. The tuned DNN model was utilized to predict the absorbance contributions, and the determined mole fractions came out to be 132, 261, 419, 6, 160, and 780 ppm of the target species, respectively. The predicted mole fractions were multiplied by the reference absorbance spectra

from PNNL (Sharpe et al., 2004) to obtain the individual absorbance spectra of BTEX species shown in Fig. 6. These were added to reproduce the composite absorbance spectrum (dashed line), which turned out to be in good agreement with the measured spectrum (solid line), as shown in Fig. 6.

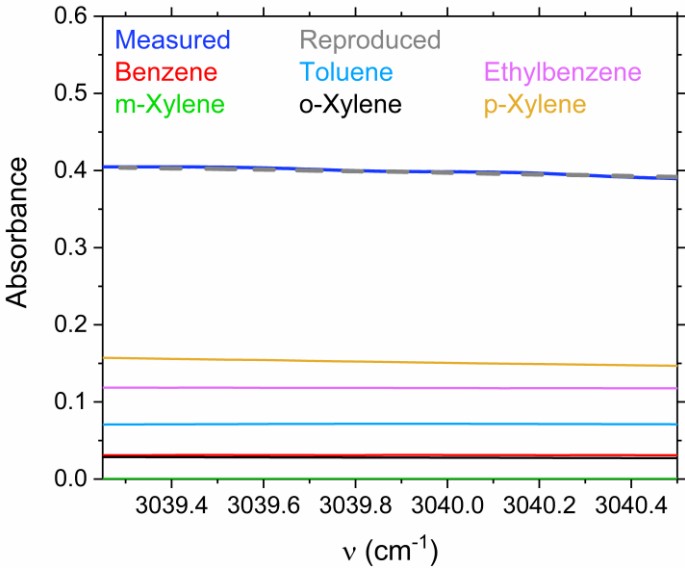

**Figure 6: Measured and reproduced absorbance spectra of a mixture containing 121, 278, 422, 0, 151, and 775 ppm of benzene,**
**toluene, ethylbenzene, m-xylene, o-xylene, and p-xylene, respectively. The individual absorbance spectra were determined by multiplying the predicted mole fractions by the reference absorbance spectra from PNNL (Sharpe et al., 2004).**

Figure 7 plots experimentally determined mole fractions of BTEX against manometric (known) values. The dotted line represents the linear fit, which demonstrates remarkable agreement between the measured and manometric mole fractions. The residuals of these concentrations are shown in the bottom panel of Fig. 7. For all species, the residual is within 10% of the

reading, which is in good agreement with the uncertainty of the measurements calculated using Eq. (5). The higher residual at low mole fractions (< 300 ppb) is due to the multi-fold dilution process used for preparing the mixtures to reach such low mole fraction values.



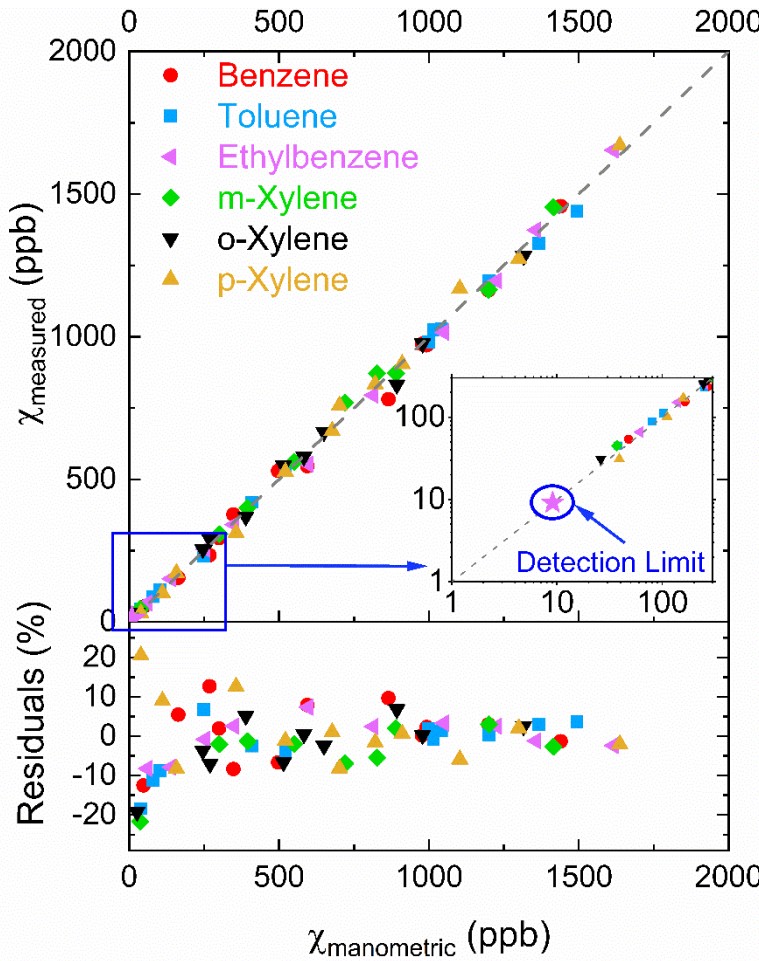

**Figure 7: Comparison of experimentally-measured and manometric mole fractions of BTEX species at ambient conditions. The**
**dashed line represents the linear fit. The bottom panel shows the residuals between the measured and manometric mole fractions.**

**4.3 Minimum detection limit**

The minimum detection limit (MDL) is one of the most important parameters in ultra-sensitive sensors (Tittel et al.). While
this can be obtained from the Allan variance of the absorbance signal, a more reasonable approach is to determine it from the
signal-to-noise-ratio (SNR) of the measured signal, corresponding to an SNR = 1. Signal noise is reflected as fluctuations of
the detected signal, which arise from various sources, such as thermal noise of the detector, laser intensity fluctuation,
mechanical oscillations of the sensor components, and optical interference/reflection from the mirrors/lenses. The latter is the
dominant noise factor in the case of multi-reflection schemes used for high sensitivity. Figure 8 shows two incident laser
intensities through a non-absorbing (vacuum) medium. Ideally, both signals should perfectly overlap, resulting in an
absorbance value of zero. However, due to signal fluctuations/noise, a minimum measurable absorbance was determined to be



~ 0.1%, which corresponds to an SNR of 1. Equation 2 is then used to convert this absorption to minimum detection limits of 11.4, 9.7, 9.1, 10, 15.6, and 12.9 ppb for benzene, toluene, ethylbenzene, m-xylene, o-xylene, and p-xylene, respectively.

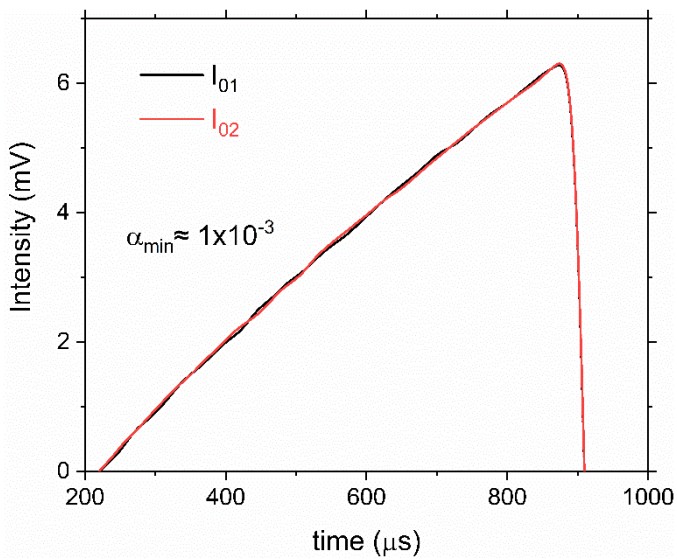

**Figure 8: Two representative incident laser intensity signals ($I_0$) showing the fluctuations in these signals.**

## 5 Conclusions

A laser sensor based on cavity enhanced absorption spectroscopy has been demonstrated to measure trace concentrations of BTEX species. The single-pass absorption increased by more than three orders of magnitude due to the high reflectivity of the mirrors, thus enabling BTEX detection down to 9.1 ppb. This detection limit is obtained at an optimum integration time of 10 seconds. Wavelength tuning and deep neural networks were utilized to enable simultaneous and selective BTEX measurements. The sensor performance was validated with manometrically prepared BTEX mixtures. The sensor can be

applied to monitor air quality in petrochemical facilities where the emissions of these species are likely, and near residential and industrial areas to monitor occupational/industrial hygiene and air pollution.

*Code availability.* Code can be provided on request.

*Data availability.*    Data can be provided on request.

*Author contributions.* MM wrote the manuscript, developed the codes, prepared the experimental setup, ran experiments, and processed the data. MS helped in running the experiments and developing the models. AA and JL helped with the initial concept/design of the sensor. AF supervised the project. All authors reviewed the manuscript.




*Competing interests.* The contact author has declared that neither they nor their co-authors have any competing interests.

*Acknowledgements.* This work was funded by the Environmental Protection Department (EPD) of Saudi Aramco (RGC/3/4749-01).

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
