# Peer review of "A Highly Sensitive and Selective Laser-Based BTEX Sensor for Occupational and Environmental Monitoring"

_EGUsphere, 2023_

## Author Comment (AC2)

We thank the reviewer for his/her constructive comments which allowed us to improve our manuscript. Our replies to the comments are given below:

Although the authors have an interesting subject for a scientific study, the manuscript is not convincing and lacks scientifically too many flaws to be accepted for publication.

- The discussion on the cavity enhanced absorption (line 87 to 99) is confusing and not according the generally agreed formulae. I suggest to follow the formulas in the article: Cavity-enhanced absorption spectroscopy of molecular oxygen by the Gianfrani et al., Journal of the Optical Society of America B Vol. 16, Issue 12, pp. 2247-2254 (1999) https://tf.nist.gov/general/pdf/1324.pdf There is also shown that the increase in path length (or absorption) is equal to π.sqrt(R)/(1-R)

While the paper suggested by the reviewer (published in 1999) is indeed a good reference on path-length amplification, our discussion is also well-established and follows the book by Gagliardi and Loock (2014) [1]. In fact, both references and discussions therein are similar and lead to the same results. Following Gianfrani's equations [2]:

$$F = \frac{\pi\sqrt{r_1 r_2}}{1 - r_1 r_2} = 4336$$

where $r_1 = r_2 = 99.964\%$. Thus, for the path-length of 5 cm given in manuscript, the equivalent pathlength is:

$$L_{eq} = \frac{2}{\pi} F L = 138.88 \, m$$

This is same as the effective pathlength (139 m) given in line 157 of the manuscript based on our approach.

- Figure 1 is not convincing concerning an interference free and selective sensing. I do not see the advantage compared to the UV. Where are the fingerprint spectra of the gases next to their absorption strengths? How orthogonal are the gases to each other?

In the UV, most hydrocarbons have broad overlapping spectra which makes it very challenging to isolate BTEX from other interfering molecules, whereas in the selected wavelength region, the main absorbers are only BTEX. Another advantage of the IR region versus UV is the availability of small-form semiconductor lasers in the IR region. Indeed, the goal of Fig. 1 is not to show the feasibility of interference-free and selective sensing of BTEX near 3.3 μm, but rather to show their overlapping spectra as an opportunity for their simultaneous sensing using DNN (deep neural networks). The absorbance spectra of BTEX are non-orthogonal in the selected wavelength range, which obfuscates their selective sensing using traditional fitting algorithms.

- The discussion on the proper selection of the spectral wavelength is not convincing. Show the broadband spectra (between 3000 and 3100 cm$^{-1}$), next to the spectra of water vapor and $CO_2$. The latter two are important because their concentrations are orders of magnitude higher.

We have added 400 ppm $CO_2$ and 1.5 % water vapor (typical mole fractions in air) spectra to the selected wavelength range to show their low absorbance in comparison to the target BTEX species. We only showed these spectra in the narrow selected range rather than 3000 – 3100 cm$^{-1}$ because water absorbance is much higher than the target species in that range and it will overwhelm their spectra. The most important thing is that in the selected region (3039.25 – 3040.5 cm$^{-1}$), water and $CO_2$ absorbance is much smaller than BTEX.

- Since BTEX consist of 6 molecules, give all 6 bands.

We have added these to the manuscript.

- Line 124: The focusing lens is not described. How is the beam divergence of the ICL? What was the type and performance of the oscilloscope?

We have added these to the manuscript.

- Line 127-128 give a reference

We have added a reference.

- Line 140: A BTEX mixture was used containing 6 gasses. What was the uncertainty in the concentration of each of the gases in the mixture? It seems here that all the gases are in one mixture, although line 153 states something else…?

Uncertainties are now stated for each BTEX molecule. In Allan variance and sensor validation measurements, the target gases were mixed to prepare one mixture. However, in the cavity gain measurements, each species was diluted separately in nitrogen to characterize the gain with respect to the reference absorption cross-section of each species. This is because the magnitudes of absorption cross-sections are different for each molecule.

- A 99.97% reflectivity of the mirrors means an enhancement of 10470 in path length (see reference). With a cavity length of 5 cm, the total path length should be 520 meter. In the manuscript the Gain is 2800; can these two be compared? How is the overall NEAS (Noise Equivalent Absorption Sensitivity) of the system?

Based on Eq. (5) in the reference provided by the reviewer [2], the enhancement factor is:

$$\frac{2}{\pi}F = \frac{2}{\pi}\frac{\pi\sqrt{r_1 r_2}}{1 - r_1 r_2} = \frac{2\sqrt{r_1 r_2}}{1 - r_1 r_2}$$

This means that a 99.97% reflectivity should give an enhancement of 3333 (not 10470). However, a reflectivity of 99.964% (as indicated in the manuscript) should give an enhancement factor of 2777 (according to the provided reference [2]). This is in perfect agreement with the gain value obtained in the manuscript using our approach (2789).

The NEAS (Noise Equivalent Absorption Sensitivity) is 0.01%, as shown in the bottom panel of Fig. 9 of the revised manuscript.

- What was the averaged transmitted power through the cavity? How is this compared to the detector sensitivity and where is the performance of the latter in the Allan curve (Fig.3)?

The average transmitted power through the cavity is calculated according to the following equation [3]:

$$I_t \approx I_0 \frac{1}{2}(1 - R) \approx 0.36 \, \mu W$$

This is well above the noise-equivalent-power (NEP) of the photodetector, which is calculated as follows:

$$NEP = \frac{\sqrt{A\Delta f}}{D^*} = 40 \, pW$$

where $A = 4 \, mm^2$ is the area of the photosensitive region, $\Delta f = 200 \, MHz$ is the bandwidth, and $D^* = 7 \times 10^{11} \, cm\sqrt{Hz}/W$ is the specific detectivity of the photodetector.

The Allan variance curve of the photodetector signal is given in Fig. R1. We have updated Figure 3 in the manuscript with this one.

[Figure]

*Figure R1. Allan variance plots of the photodetector signals.*

- The spectra leading to Fig.4 are not shown, while these are essential to convince the interested reader. Furthermore, I do not see any error bars in the figure.

Error bars are already included in the figure (new Fig. 5). They are clearer in the high absorbance region (top right corner) because of the higher error.

The spectra leading to Fig. 4 are based on absorbance of single species, where each BTEX was diluted separately in nitrogen to obtain the cavity gain, as mentioned in the manuscript. A sample of these spectra is now added to the manuscript as Fig. 4.

- Equation 11: Why is k limited from 1-4 and not extended to 6 (of higher, due to interfering species)?

Thank you for pointing this out, it was a typo and now we fixed it.

- While the uncertainty of alpha(CEAS) was determined to be 2% (line 160), the mixtures were calculated to be 3-digit accuracy (line 210). There is no convincing performance of the DNN model to be justified.

Uncertainties are now added to the measured mole fractions (3.76%) based on combining the RMSE of the DNN model and the uncertainty in the measured cavity absorbance.

- Fig 7 cannot be properly judged due to all the raised questions above

Now that the above concerns have been clarified, we hope that Fig. 7 is clear.

- Fig 8 is not convincing.

We have now added the absorbance resulting from the fluctuation in the two incident laser intensity signals to make it more convincing.

**References:**

[1] Gagliardi, Gianluca, and Hans-Peter Loock, eds. *Cavity-enhanced spectroscopy and sensing*. Vol. 179. Berlin: Springer, 2014.

[2] Gianfrani, Livio, Richard W. Fox, and Leo Hollberg. "Cavity-enhanced absorption spectroscopy of molecular oxygen." *JOSA B* 16.12 (1999): 2247-2254.

[3] Moyer, E. J., et al. "Design considerations in high-sensitivity off-axis integrated cavity output spectroscopy." *Applied Physics B* 92 (2008): 467-474.